# Multi-Site Photoplethysmographic and Electrocardiographic System for Arterial Stiffness and Cardiovascular Status Assessment

**DOI:** 10.3390/s19245570

**Published:** 2019-12-17

**Authors:** David Perpetuini, Antonio Maria Chiarelli, Lidia Maddiona, Sergio Rinella, Francesco Bianco, Valentina Bucciarelli, Sabina Gallina, Vincenzo Perciavalle, Vincenzo Vinciguerra, Arcangelo Merla, Giorgio Fallica

**Affiliations:** 1Department of Neuroscience and Imaging, Institute for Advanced Biomedical Technologies, University G. D’Annunzio of Chieti-Pescara, Via Luigi Polacchi 13, 66100 Chieti, Italy; antonio.chiarelli@unich.it (A.M.C.); sgallina@unich.it (S.G.); arcangelo.merla@unich.it (A.M.); 2STMicroelectronics, ADG R&D, Stradale Primosole 50, 95121 Catania, Italy; lidia.maddiona@st.com (L.M.); vincenzo.vinciguerra@st.com (V.V.); piero.fallica@gmail.com (G.F.); 3Physiology Section, Department of Biomedical and Biotechnological Sciences, University of Catania, Via Santa Sofia 97, 95123 Catania, Italy; sergio.rinella@hotmail.com (S.R.); perciava@unict.it (V.P.); 4Institute of Cardiology, University G. D’Annunzio of Chieti-Pescara, Via Dei Vestini 5, 66100 Chieti, Italy; dr.francescobianco@gmail.com (F.B.); valentina.bucciarelli@unich.it (V.B.); 5Kore University, Department of Sciences of Life, Viale delle Olimpiadi, 94100 Enna, Italy

**Keywords:** photoplethysmography (PPG), electrocardiography (ECG), pulse wave velocity (PWV), cardiovascular status, silicon photomultipliers (SiPMs)

## Abstract

The development and validation of a system for multi-site photoplethysmography (PPG) and electrocardiography (ECG) is presented. The system could acquire signals from 8 PPG probes and 10 ECG leads. Each PPG probe was constituted of a light-emitting diode (LED) source at a wavelength of 940 nm and a silicon photomultiplier (SiPM) detector, located in a back-reflection recording configuration. In order to ensure proper optode-to-skin coupling, the probe was equipped with insufflating cuffs. The high number of PPG probes allowed us to simultaneously acquire signals from multiple body locations. The ECG provided a reference for single-pulse PPG evaluation and averaging, allowing the extraction of indices of cardiovascular status with a high signal-to-noise ratio. Firstly, the system was characterized on optical phantoms. Furthermore, in vivo validation was performed by estimating the brachial-ankle pulse wave velocity (baPWV), a metric associated with cardiovascular status. The validation was performed on healthy volunteers to assess the baPWV intra- and extra-operator repeatability and its association with age. Finally, the baPWV, evaluated via the developed instrumentation, was compared to that estimated with a commercial system used in clinical practice (Enverdis Vascular Explorer). The validation demonstrated the system’s reliability and its effectiveness in assessing the cardiovascular status in arterial ageing.

## 1. Introduction

Photoplethysmography (PPG) [1] is a noninvasive optical technique that measures local blood volume changes caused by cardiac pulse propagation into the arterial tree. This technique is an active technology that requires a source to inject light into the skin and a detector to collect photons after their propagation in biological structures. Near-infrared (NIR) light is commonly used for PPG because of its sensitivity to oscillating hemoglobin concentration and low light absorption of tissues in this spectral range, that therefore allows the investigation of deep structures (a few cm) [2]. The PPG signal consists of a baseline component that depends on blood volume and a modulated component at around 1 Hz caused by pulse propagation. Pulse propagation induces a periodic modulation of arterial size resulting in local changes of hemoglobin concentration that can be assessed optically. PPG measurements can be performed in transmission modality (e.g., on fingers to evaluate arterial oxygen saturation and pulse wave contours [3]) or, by exploiting the highly diffusive properties of biological tissue in the NIR spectral range, in back-reflection modality [4]. In the back-reflection configuration, the light source and the detector are placed on the same surface at an interoptode distance of few centimeters. This configuration allows the investigation of thick structures where transmission modality is not feasible and the interrogation of large-caliber vessels such as brachial and tibial arteries. PPG is used in commercial devices to evaluate arterial oxygen saturation, heart rate, heart rate variability, and arterial compliance [1]. Moreover, PPG can be employed for cardiovascular status assessment relying on other metrices derived from peculiar features of the pulse wave contour [5,6,7,8]. A common metric employed in clinical practice is pulse wave velocity (PWV). PWV is positively associated with arterial stiffness [9,10], being roughly proportional to the square root of the incremental elastic modulus of the vessel wall where the pulse propagates [11]. Generally, PWV estimation requires measurements at two different body locations, for example the brachium and ankle (baPWV). PWV can also be estimated by performing PPG at one body location, providing a measurement of the distance from the heart and a temporal reference to the heart pulsation [12]. For example, the reference can be acquired through electrocardiography (ECG) by lockage to specific ECG voltage peaks [13,14]. ECG is a surface measurement of the electrical potential produced by electrical activity in the cardiac tissue. Among distinct ECG waves, the QRS complex represents ventricular depolarization. Generally, for combined PPG–ECG measurements, the R peak is employed as a locking point for average PPG signal estimation.

The development and validation of an innovative multi-site PPG–ECG system is presented here. The system enables accurate evaluation of cardiovascular and arterial status. It was developed within an H2020 project funded by the European Union (termed ASTONISH). The system is suited to use in medical and pre-clinical environments. It overcomes the limitations of devices commercially available for arterial stiffness monitoring through the integration of a large number of wearable, mechanically stable, and fully synchronized optical and electrical probes.

Thanks to the employment of sensitive and large-area detectors which maximize photon harvesting from the skin, namely n-on-p silicon photo-multipliers (SiPMs) [15], PPG signals can be collected at large source–detector distances, effectively investigating large arteries and robustly assessing the cardiovascular status with a high signal-to-noise-ratio (SNR). SiPMs are bi-dimensional arrays of pixelated semiconductor detectors made of single-photon avalanche diodes (SPADs) working in Geiger Mode [16]. Compared to other semiconductor detectors, SiPMs present major advantages of sensitivity, high internal gain, and speed of response [17]. SiPMs provide a much higher responsivity, of ~3 to ~5.5 orders of magnitude [18], than standard photodiodes or avalanche photodiodes (APDs). Previous work in other biomedical applications showed SiPMs linearity with light sensitivity down to a few femtoWatts (fW) [15,19]. Compared to other sensitive detectors, such as photomultiplier tubes (PMTs), SiPMs are much more compact, easier to handle, mechanically robust, optically resistant, and electrically reliable and operate at a much lower voltage.

The system’s SNR was further improved through ambient light suppression based on light filters in the sensor technology and ad hoc readout electronics.

The system was firstly characterized on optical phantoms by evaluating the extent of cross-talk among PPG probes. Secondly, the system performance was assessed in vivo, by estimating the intra- and extra-operator repeatability of the baPWV and its association with age. Moreover, the baPWV estimates were compared to those obtained using commercial technology.

## 2. Materials and Methods

### 2.1. Optical Probes and Electrodes

Figure 1a shows a manufactured PPG optical probe that was used in the developed system. Each probe was made of a light-emitting diode (LED) and a SiPM mounted on the same board.

The n-on-p SiPMs utilized were fabricated at STMicroelectronics (Catania, Italy) and they were packaged in a surface mount device (SMD) of 5.1 × 5.1 mm^2^ area. They featured 4871 square microcells of 60 µm pitch, covering a total area of 4.18 × 4.68 mm^2^ with a geometrical fill factor of 67.4%. The SiPMs were manufactured employing heavily doped p-type silicon substrates. The diode anode was composed of a thin p-type epitaxial layer grown on the wafer and a very heavily doped, very thin, n-type polysilicon layer added by high-temperature chemical vapor deposition. The doping profile of SPAD diodes was n^++^/p^−^/p^+^, allowing for high sensitivity in the NIR spectral range. The quenching resistor was made of low-doped polysilicon and was integrated inside the cell. Moreover, thin optical trenches were put around the microcell active area in order to reduce the cross-talk effect between adjacent microcells [20].

The SiPMs were equipped with an optical cast plastic IR long-pass filter (Edmund Optics Inc., Barrington, NJ, USA) characterized by optical transmittance higher than 90% for wavelengths equal to or higher than 940 nm. The filter was placed on the SMD package using 352TM adhesive (Loctite®, Milan, Italy) effectively absorbing more than 99% of environmental light below 650 nm.

An SMC940 LED (Roithner LaserTechnik, Vienna, Austria) based on Gallium Arsenide(GaAs) technology, assembled in the SMD package and emitting at 940 nm, was used as a light source. The LED had an area of 3 × 2 mm^2^, a viewing angle of 110°, a spectral bandwidth of 100 nm, and power emission up to few mW.

The SiPM and LED board were inserted in bracelets equipped with pressurized cuffs (Figure 1b) that, delivering a pressure below that of diastole (~60 mmHg), ensured an optimal coupling of the optical probes with the skin without altering the shape of the pulse. All the PPG probes employed a back-reflection recording configuration with an interoptode distance of 2 cm (Figure 1b). 

The ECG signal was acquired through disposable F2080 ECG electrodes from FIAB (Vicchio, Italy). These electrodes were equipped with an Ag/AgCl electrode and stainless-steel clips. A small amount of gel was used to stabilize contact with the skin and to reduce impedance. 

### 2.2. System Architecture

A printed circuit board (PCB) [21,22] (Figure 2a) was developed to interface the PPG and ECG probes with an NI PXIe4303 (National Instruments, Austin, TX, USA) analog input module (Figure 2b). The acquisition module featured 32 differential analog inputs and analog-to-digital converters (ADCs), one per each channel, with a resolution of 24 bits. The ADCs’ input full scale was set at 10 V for the 6 PPG channels and at 100 mV for the 4 ECG channels. The PPG and ECG signals were both acquired with a sampling frequency of 1 kHz.

The PCB board was equipped with a 4 V portable battery, power management circuits, a conditioning circuit for the SiPMs signals, 8 mini B-USB connectors for PPG probes, and 8 SMA output connectors. A voltage regulator was set at 3.3 V to provide power supply. In the same board, a step-up DC–DC converter generated an output of 30 V and provided a bias to the SiPMs. Trimmers on the PCB allowed adjustment of the LEDs optical power. 

A LabVIEW software program was developed to acquire PPG and ECG signals. The software controlled the 24-bit ADC NI PXle-4303 NI acquisition system. A graphical user interface (GUI) (Figure 2c) implemented in the NI LabVIEW environment allowed management of the PPG and ECG signals. The GUI displayed the filtered PPG signal, its first and second derivatives, and the ECG signal. The program allowed us to compare two PPG signals acquired from probes arranged at different body sites and measure their time-delay. After signal acquisition, the overall dataset was stored in a log file.

The final system could acquire up to 8 PPG probes that could be located along the course of principal arteries on each hemibody (e.g., carotid, brachial, radial/ulnar, and tibial arteries) and 10 ECG electrodes that could acquire 12-lead ECG (i.e., 4 registration locations on the limbs [right and left arm, right and left foot] and 6 registration locations in the precordial area [V1, V2, V3, V4, V5, V6]). Figure 3 reports a schematic representation of the ECG and PPG probes’ placement on a body template. The system was tested employing 6 PPG probes and 3 ECG electrodes. The PPG probes were placed on the brachial, radial/ulnar, and tibial arteries for the two hemibodies. The 3 ECG electrodes were placed on the patient’s body following the standard Einthoven’s triangle lead configuration [23].

### 2.3. Preprocessing of PPG and ECG Signals

Raw PPG signals were converted into optical densities (ODs) [24]. Raw ODs and ECG signals were filtered by employing band-pass, 4th order, zero-lag, Butterworth digital filters. For PPG, the cut-off frequencies were set at 0.2 and 10 Hz; for ECG, the cut-off frequencies were set at 0.2 and 30 Hz. The ECG R-wave peaks were identified considering the local maxima on a filtered and normalized (z-score) Lead I ECG signal, defining some constraints (minimum value of the peak: 3; interpeak distance: 600 ms). The procedure provided an excellent automatic R peak identification without the need of any manual correction after visual inspection.

PPG average pulse was estimated in all the optical channels available in a time window from 0.3 s prior to 1.2 s after the R-wave peak. The amplitude of the time window of 1.5 s allowed the investigation of pulsation frequencies down to a minimum value of 40 beats/min. In order to exclude possible noisy periods of PPG signal, a trim-mean approach for the average signal evaluation was applied by excluding single pulses with at least a value below or above the 25th or 75th percentile of the sample population. Thanks to the high quality of the signals and the reduced presence of motion artifacts, this analysis guaranteed a reliable estimation of the pulse average PPG with small standard errors. The ECG and PPG preprocessing chain and an example of PPG average pulse and standard error for each time point are reported in Figure 4.

### 2.4. System Characterization and Validation

As a first step of system characterization, the cross-talk amongst the PPG probes was estimated on optical phantoms. One optical probe was firstly kept still for 20 s and after it was moved with a frequency of 1 Hz on an optical phantom, while a second probe was placed on a control optical phantom, and a third one was positioned on the ulnar artery of a subject. The effect of the motion artifacts was tested on the probes through temporal signal and Fourier-transform visual inspection. 

The in vivo validation of the system was performed in agreement with the ethical standards of the Helsinki Declaration, 1964, and approved by the Ethical Committee Catania 1 (authorization n. 113/2018/PO). All subjects involved, after having been informed about finalities and methodologies of the study, provided written informed consent and could withdraw from the experiment at any time.

The in vivo validation of the system assessed the repeatability of the baPWV estimation. Ten healthy participants were enrolled for the experiment. Each participant was asked to lay supine on a medical cot. Two operators placed the probes on the different measurement sites, and each of them repeated the measurements twice in order to test both the intra- and extra-operator repeatability. baPWV was computed from pulse average PPG signals. In particular, PWV was estimated on the basis of the time delay of the diastolic peak (foot) of the PPG signal [5], and the geometrical distance between the brachial and the ankle PPG probes for both sides were evaluated. PWV repeatability was assessed on left and right average values using Bland–Altman plots [25,26,27], paired t-tests [28], and correlation analysis [29].

Moreover, in order to test the ability of the system to estimate parameters indicative of vascular ageing, 78 healthy participants with a large age range (from 20 to 80 years of age) were enrolled in the study. Cross-sectional correlation between participants’ age and left and right baPWV was estimated, exploiting the known monotonic associations between age, arterial stiffness, and PWV [30]. Finally, the performance of the system in estimating the baPWV was assessed through comparison with a commercial system, namely, the Enverdis Vascular Explorer (VE) (Düsseldorf, Germany) [31,32]. A Bland–Altman plot, paired t-test, and correlation analysis were performed to test the consistency of the two measurements. Figure 5 shows an in vivo measurement during the VE evaluation (Figure 5a) and ECG–PPG signal acquisition (Figure 5b). It is worth noticing that VE had only two probes, hence the left and right sides of the body had to be investigated separately, increasing the examination duration.

## 3. Results

Figure 6 reports an example of results of the evaluation of PPG probes cross-talk. The absence of cross-talk between different PPG channels is visible. No motion artifact, which is clear on the optical probe that was moved, can be observed on the concurrently acquired probe located on the control phantom and the probe located on the subject, both in the time (Figure 6a) and in the frequency (Figure 6b) domain. Similar results were obtained for all possible arrangements of the PPG probes.

Figure 7 reports an example of single-pulse averaged PPG signals (locked to the R peak of the ECG), with associated standard error, concurrently acquired from the different body locations for two indicative participants of the 10 subjects, for the repeatability measurements. It is possible to note the small intra-subject variability of the signal (low standard error) associated with an evident inter-subject difference.

To further evaluate the intra-subject and intra-session stability of the PPG signal, the single-beat repeatability was evaluated through a correlation analysis. A minimum correlation between single beats of 0.94 was found among the participants, further corroborating the high SNR and stability of the measurement. 

Figure 8 reports the intra- (Figure 8a,b) and extra-operator (Figure 8c,d) average left and right baPWV repeatability analysis performed on 10 subjects. Figure 8a,c are correlation plots, whereas Figure 8b,d are Bland–Altman plots. For the intra-operator measurements, the root-mean-square error (RMSE) of the velocity was 0.35 m/s, the correlation coefficient was 0.93, and the paired t-test did not show significant differences between the two measurements (t(9) = 0.17; p = 0.86). For the extra-operator repeatability, the RMSE was 0.35 m/s, the correlation coefficient was 0.94, and the paired t-test was not significant (t(9) = −0.09; p = 0.93).

Figure 9 reports the correlation between age and baPWV for a cohort of 78 healthy participants. A good correlation was found for both the right side (r = 0.85, Figure 9a) and the left side of the body (r = 0.80, Figure 9b).

Figure 10 reports a comparison between VE and ECG–PPG in estimating baPWV. The correlation coefficients were 0.70 (Figure 10a) and 0.78 (Figure 10c) for the right and left side of the body, respectively. The Bland–Altman plot showed no proportional errors in baPWV estimation from PPG with respect to VE (right body side, Figure 10b; left body side, Figure 10d). Paired t-tests showed a significant average difference between the two techniques, highlighting a bias for the PWV estimated from PPG (right body side: t(77) = −7.2; p = 3.2∙10^−10^, difference mean = 2.4 m/s); left body side: t(77) = −5.9; p = 8.8∙10^−8^, difference mean = 1.5 m/s).

## 4. Discussion

An innovative multi-channel PPG–ECG system was presented. The PPG–ECG system could be equipped with up to 8 PPG probes and 10 ECG electrodes, and was tested employing 6 PPG probes, enclosed in bracelets equipped with pressure cuffs and 3 ECG electrodes. The six PPG probes were placed on the left and right brachial artery, left and right radial/ulnar artery, and left and right tibial artery. A high sensitivity of the probes was obtained by employing SiPMs thanks to their ability to detect single photons with a large detection area, allowing the collection of signals from deep vessels such as brachial and tibial arteries. PPG and ECG recordings were synchronized using a dedicated PCB that, together with a GUI implemented in the NI LabVIEW environment, ensured control of the 24-bit acquisition system. The system showed no visible cross-talk between the different PPG channels and good intra- and extra-operator repeatability when evaluated on baPWV estimates. Moreover, the correlation between age and baPWV confirmed the capability of the PPG–ECG system to assess cardiovascular status in ageing. This result was further confirmed by comparison with a commercial system that evaluates baPWV (VE). In fact, the baPWV estimated from PPG was highly correlated with the baPWV estimated by VE, with a correlation coefficient of r = 0.70 for the right side of the body and of r = 0.78 for the left side of the body. The Bland–Altman plot showed a good consistency between the two estimations. Moreover, systematic or proportional errors were absent, as shown by the Bland–Altman plot. However, paired t-tests highlighted a significant average difference between the two techniques of around 2 m/s. This difference could be due to the different types of measurement employed. In fact, VE is based on pressure sensors, whereas PPG measures volumetric oscillations; hence, the difference in the estimated PWV could be caused by a pressure–volume hysteresis [33,34]. In fact, differences in absolute values for parameters estimated from pressure and volumetric sensors have already been found by several researchers [7,35]. Another possible explanation of this effect could be associated with a systematic differential error between instrumentations in the estimation of the probes’ distance. 

The developed PPG–ECG system had the ability to perform back-reflection recordings at large interoptode distance, allowing the evaluation of PPG signals on large arteries. Moreover, when compared to commercially available devices, the system had the advantage of employing a large number of probes. This indeed allowed synchronous monitoring at multiple body locations. Although in the current study we analyzed a small subset of the recorded locations separately, multisite-PPG could be employed to evaluate coupled or synergistic effects among body locations indicative of cardiovascular status. These effects could be highlighted by employing data-driven multivariate analysis (e.g., through linear approaches such as using a general linear model [GLM] approach) [36,37,38].

Importantly, this analysis might be performed with a simple and short recording session. The brevity of the recordings is a crucial feature in clinical applications, fostering the introduction of this technique in large-scale arterial stiffness and cardiovascular risk screening [39].

## 5. Conclusions

The development and validation of a multi-site PPG–ECG system working in a back-reflection recording modality were presented. The optical probes did not show cross-talk effects, ensuring a good quality of the recorded signals. The baPWV, estimated from PPG signals, showed high repeatability and, as expected, was sensitive to ageing and was consistent with baPWV obtained employing a commercial pressure sensor device (Vascular Explorer). The system’s capabilities to concurrently measure multiple body locations with simplicity and in a short recording time may pave the way to multivariate approaches for arterial stiffness and cardiovascular ageing assessment through PPG and ECG. Thanks to the technological development presented, ECG–PPG systems could become a common tool in research and clinical settings for cardiovascular evaluation. 

## Figures and Tables

**Figure 1 sensors-19-05570-f001:**
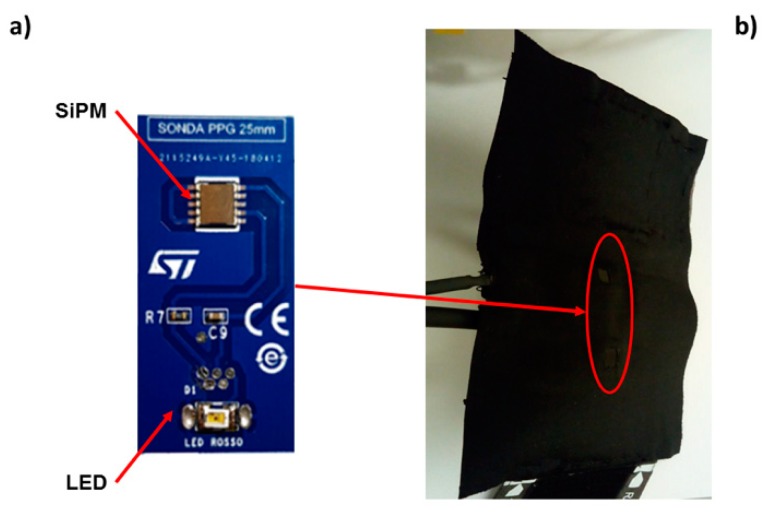
Photoplethysmography (PPG) optical probe, manufactured at STMicroelectronics (Catania, Italy), that was used in the developed system. The probe worked in a back-reflection recording modality. (**a**) Each probe was made of a light-emitting diode (LED) and a silicon photo multiplier (SiPM) mounted on the same board. (**b**) The SiPM and LED board were inserted in bracelets equipped with pressurized cuffs delivering a pressure below that of diastole (~60 mmHg).

**Figure 2 sensors-19-05570-f002:**
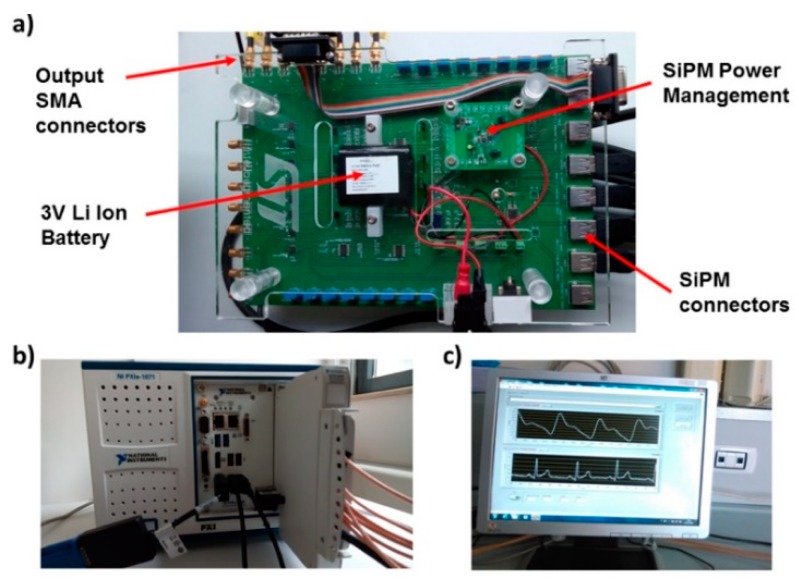
(**a**) Printed circuit board (PCB) developed to interface PPG probes and electrocardiography (ECG) electrodes with the analog input module. (**b**) NI PXIe4303 (National Instruments, Austin, TX, USA) analog input module that was used in the system. (**c**) LabVIEW graphical user interface (GUI) program that acquired and allowed the visualization of PPG and ECG signals.

**Figure 3 sensors-19-05570-f003:**
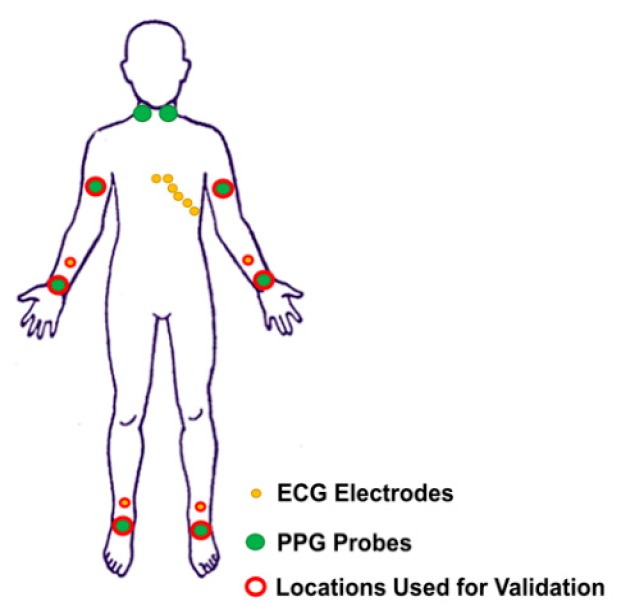
Schematic representation of the acquirable ECG and PPG locations reported on a body template.

**Figure 4 sensors-19-05570-f004:**
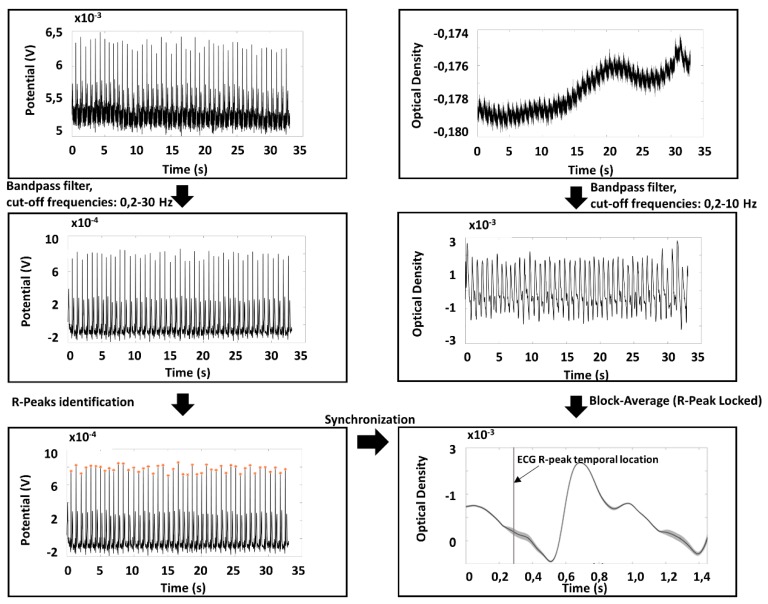
ECG and PPG preprocessing chain used for data analysis together with an example of an extracted PPG average pulse and associated standard error.

**Figure 5 sensors-19-05570-f005:**
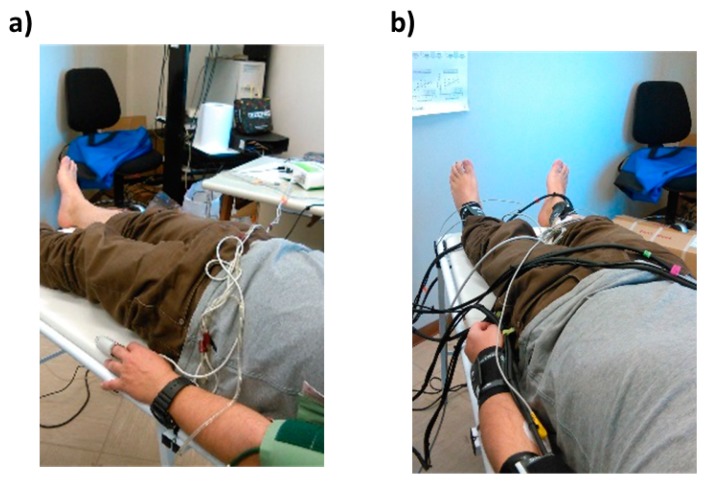
(**a**) Example of in vivo measurement using Vascular Explorer (VE) and (**b**) example of ECG–PPG signal acquisition using the developed system.

**Figure 6 sensors-19-05570-f006:**
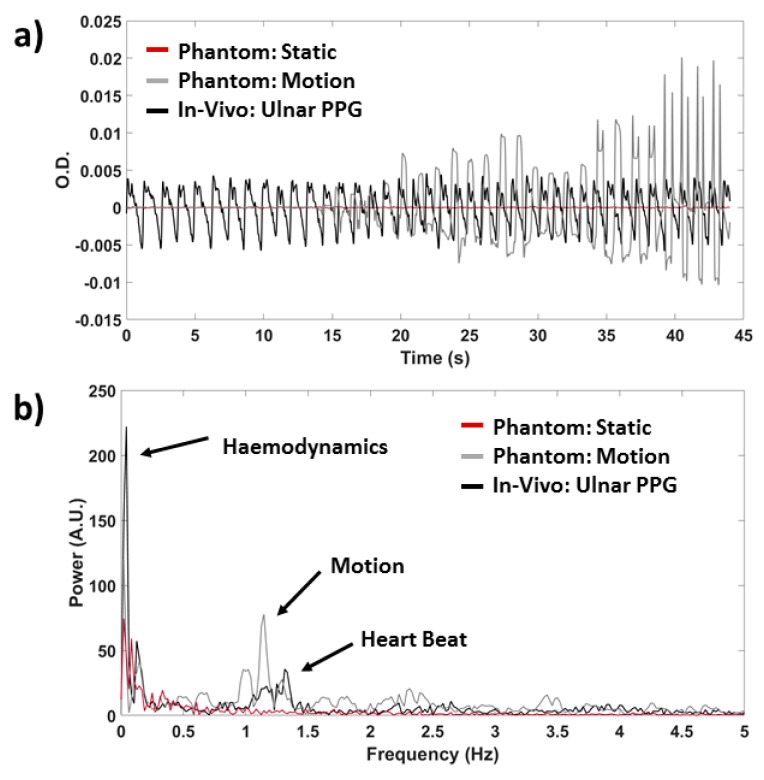
Example of cross-talk results for three randomly selected PPG probes (**a**) in the time and (**b**) in the frequency domain. Signals are reported for the probe subject to motion (grey), for the control measurements on an optical phantom (red) and on the ulnar artery of a subject (black).

**Figure 7 sensors-19-05570-f007:**
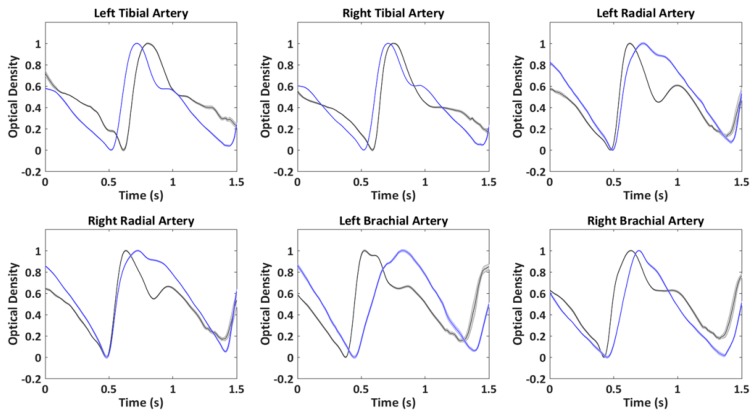
Example of single-pulse averaged PPG signals with associated standard error acquired concurrently in six body locations from two indicative participants.

**Figure 8 sensors-19-05570-f008:**
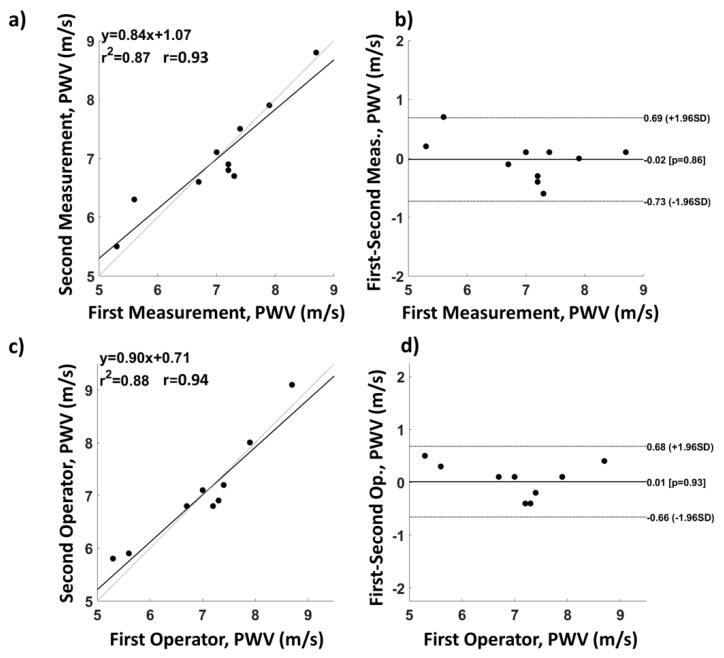
Intra- and extra-operator average left and right brachial-ankle pulse wave velocity (baPWV) repeatability analysis. (**a**) Intra-operator correlation and (**b**) associated Bland–Altman plot; (**c**) extra-operator correlation and (**d**) associated Bland–Altman plot.

**Figure 9 sensors-19-05570-f009:**
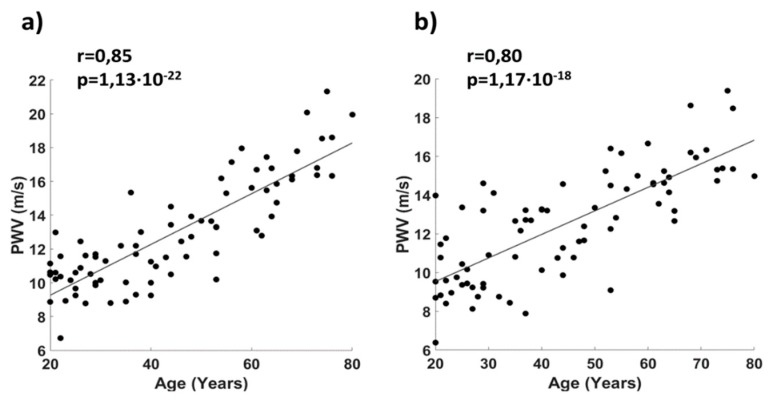
Correlation plot between age and baPWV for a cohort of healthy participants for (**a**) the right side and (**b**) the left side of the body.

**Figure 10 sensors-19-05570-f010:**
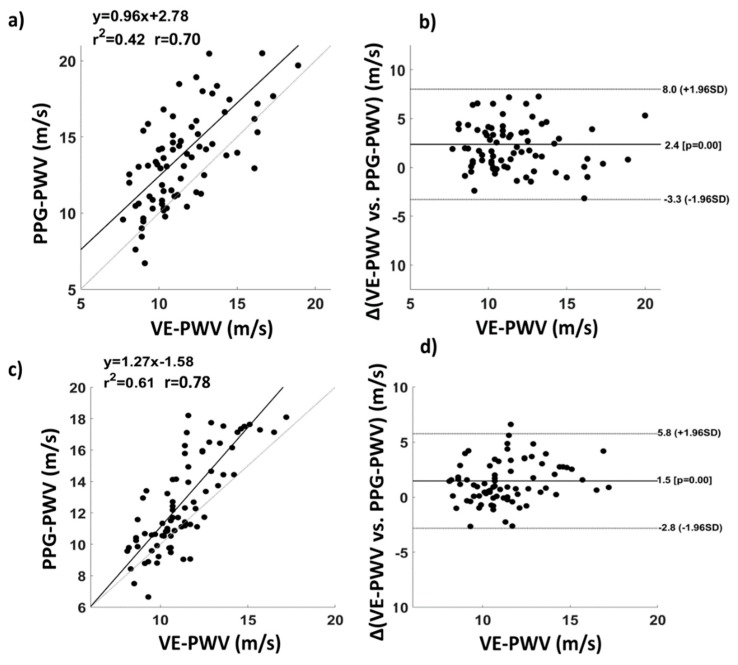
Comparison between VE and the ECG–PPG developed system in estimating baPWV. (**a**) Correlation and (**b**) associated Bland–Altman plot of baPWVs evaluated through the two instrumentations for the right side of the body; (**c**) correlation and (**d**) associated Bland–Altman plot of baPWVs evaluated through the two instrumentations for the left side of the body.

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
