# Peer review of "Multi-Site Photoplethysmographic and Electrocardiographic System for Arterial Stiffness and Cardiovascular Status Assessment"

_sensors, 2019, doi:10.3390/s19245570_

Round 1

Reviewer 1 Report

Generally interesting and well-written article.
The article has a correct and understandable organization but requires additions and corrections. Below are my comments.

1. row 22, A little incomprehensible term "hosted up".
2. row 46, 1 Hz is not fast modulation.
3. The introduction does not contain all recent publications, e.g. Simultaneous multi-site measurement system for the assessment of pulse wave delays, https://doi.org/10.1016/j.bbe.2019.01.001, The list of publications should be completed.
4. row 121, The Figure 1 should be moved up, e.g. to row 96.
5. row 151, What does "hemi body" mean?
6. Figure 3 - The location of the sensors during testing should also be marked.
7. row 179, Cutoff frequency mismatch for ECG filtration (50 Hz in the text - row 165 - and 30 Hz in the figure).
8. Figure 4 - Illegible charts. The X axis should be extended.
9. row 227, Abbreviations used in the drawing should be explained, e.g. RPC.
10. row 231, The size of the study group should be given.
11. row 232 and Figure 8a - r parameter mismatch.
12. Study group and Figure 8, Figure 9 - Do the results shown refer to the same research group? This should be clarified. If so, why are there such big differences in the maximum PWV values? (i.e. approx. 14 m/s in Figure 8 and 18 m/s in Figure 9). This should be clarified.
13. It is recommended to provide information on the bioethics commission approving the test method.

Author Response

Generally interesting and well-written article.

The article has a correct and understandable organization but requires additions and corrections. Below are my comments.

row 22, A little incomprehensible term "hosted up".

Sentence corrected to:

- The system could acquire signals from 8 PPG probes and 10 ECG leads.-

row 46, 1 Hz is not fast modulation.

Sentence modified to:

- PPG signal consists of a baseline component that depends on blood volume and a modulated component at around 1 Hz caused by pulse propagation. -

The introduction does not contain all recent publications, e.g. Simultaneous multi-site measurement system for the assessment of pulse wave delays, https://doi.org/10.1016/j.bbe.2019.01.001, The list of publications should be completed.

We want to thank the Reviewer for this comment. Some literature was added, including the one proposed by the Reviewer.

row 121, The Figure 1 should be moved up, e.g. to row 96.

The figure was moved as suggested.

row 151, What does "hemi body" mean?

Hemi body means half of the body. The term is often used to indicate the right and left part of the body divided by the sagittal plane.

Figure 3 - The location of the sensors during testing should also be marked.

We agree with the Reviewer. Figure 3 was modified as requested, highlighting the location used for the testing of the system.

row 179, Cutoff frequency mismatch for ECG filtration (50 Hz in the text - row 165 - and 30 Hz in the figure).

Thank you for spotting the error. The cut-off frequency used was 30 Hz. Corrected in line 165.

Figure 4 - Illegible charts. The X axis should be extended.

Figure 4 was corrected to make it more readable.

row 227, Abbreviations used in the drawing should be explained, e.g. RPC.

RPC stands for reproducibility coefficient. However, the metric was redundant in the figure, and it is now removed.

row 231, The size of the study group should be given.

The information is now provided:

- Figure 8 reports the correlation between age and baPWV for a cohort of 78 healthy participants. -

row 232 and Figure 8a - r parameter mismatch.

The correct r is 0.85.  The sentence is now changed as follow.

- A good correlation was found for both the right side (r=0.85, Figure 8a) and ..-

Study group and Figure 8, Figure 9 - Do the results shown refer to the same research group? This should be clarified. If so, why are there such big differences in the maximum PWV values? (i.e. approx. 14 m/s in Figure 8 and 18 m/s in Figure 9). This should be clarified.

We want to thank the Reviewer for this important observation. The results are from the same group, the figure is now corrected.

It is recommended to provide information on the bioethics commission approving the test method.

The following sentences were added at line 188:

- The in-vivo validation of the system was performed in agreement with the ethical standards of the Helsinki Declaration, 1964, and approved by the Ethical Committee Catania 1 (authorization n. 113/2018/PO). All subjects involved, after having been informed about finalities and methodologies of the study, provided written informed consent and could withdraw from the experiment at any time. -

Reviewer 2 Report

The work presents a system capable of measuring PPG and ECG on different locations of the body. We paper is well written and easy to follow and understand. Some minor editing needs to be carefully done. The work has covered system design (hardware), signal analysis, statistical benchmarking and comparison with tank measurements. Overall is more detailed and informative than usual works in the field. 

Some minor comments are the following:

- Cardiovascular risk is a specific concept and it is not covered in the scope of the work, see e.g. "Sudden Cardiac Risk Stratification with Electrocardiographic Indices - A Review on Computational Processing, Technology Transfer, and Scientific Evidence" -Frontiers 2016-. Claims about cardiovascular risk should be softened, starting with the title.

- Some graphical comparison could be made with respect to the PPG waveforms when simultaneously recorded in some patient, as well as some inter-patient variability. 

- Some additional information about waveform variability / reproducibility with time could be included, as a part of the current analysis.

Author Response

The work presents a system capable of measuring PPG and ECG on different locations of the body. We paper is well written and easy to follow and understand. Some minor editing needs to be carefully done. The work has covered system design (hardware), signal analysis, statistical benchmarking and comparison with tank measurements. Overall is more detailed and informative than usual works in the field.

Some minor comments are the following:

- Cardiovascular risk is a specific concept and it is not covered in the scope of the work, see e.g. "Sudden Cardiac Risk Stratification with Electrocardiographic Indices - A Review on Computational Processing, Technology Transfer, and Scientific Evidence" -Frontiers 2016-. Claims about cardiovascular risk should be softened, starting with the title.

We agree with the reviewer. Cardiovascular risk was replaced by cardiovascular status through the paper.

Reference to the proposed review is reported in the paper.

- Some graphical comparison could be made with respect to the PPG waveforms when simultaneously recorded in some patient, as well as some inter-patient variability.

We inserted a new Figure 7 with examples of single-pulse averaged PPG signals (with standard error) recorded from two indicative patients. The main text was integrated as follow

- Figure 7 reports an example of single-pulse averaged PPG signals (locked to the R-peak of the ECG), with associated standard error, concurrently acquired from the different body locations for two indicative participants of the 10 subjects acquired for the repeatability measurements. It is possible to note the small intra-subject variability of the signal (low standard error) associated with an evident inter-subject difference.-

- Some additional information about waveform variability / reproducibility with time could be included, as a part of the current analysis.

We added a further analysis assessing the intra-session stability of the PPG signal. We added the following sentences to the main text:

- To further evaluate the intra-subject and intra-session stability of the PPG signal, the single beat repeatability was evaluated through a correlation analysis. A minimum correlation between single beats of 0.94 was found among the participants, further corroborating the high signal to noise ratio and stability of the measurement. -